# The Private Sector as a Partner for SDG 6-Related Issues in Megacities: Opportunities and Challenges in Rio de Janeiro, Brazil

**Maria Inês Paes Ferreira** [1,*]**, Vicente de Paulo Santos de Oliveira** [1]**, Graham Sakaki** [2] **and Pamela Shaw** [2]

1. Research and Innovation Pro-Rectory—Vocational Doctoral Program in Modeling and Technology for the Environment Applied to Water Resources, Instituto Federal Fluminense, 357 Cel. Walter Kramer Street, Vera Cruz Park, Campos dos Goytacazes 28080-565, RJ, Brazil; vsantos@iff.edu.br
2. Mount Arrowsmith Biosphere Region Research Institute, Nanaimo Campus, Vancouver Island University, 900 Fifth Street, Nanaimo, BC V9R 5S5, Canada; graham.sakaki@viu.ca (G.S.); pam.shaw@viu.ca (P.S.)
* Correspondence: mferreira@iff.edu.br

**Abstract:** This article reviews recent studies that address water sustainable management opportunities and challenges in megacities around the world, with an emphasis on the case of Rio de Janeiro Metropolitan Region, one of the two megacities in Brazil. With reference to recent debates on water, megacities, and the climate crisis, as well as UN Water and Global Report Initiative documents, we focused on the implementation of the 2030 Agenda Sustainable Development Goal 6: Clean Water and Sanitation for All. The new Brazilian sanitation legal framework regulates public–private partnerships. In this context, the manuscript discusses the main question concerning water, sanitation, and hygiene that arises in the Brazilian case study: is universality possible in profit-oriented models? Through the current technical and academic literature consulted, the paper compares initiatives involving multiple stakeholder governance models that depend on private resources to implement universal access to drinking water, sanitation, and water-related extreme event controls, pointing out alternatives that can help to achieve the targets of SDG. Validation by key informants supports the synthesis of the reviewed documents, and the findings illustrate that concerted public efforts together with market mechanisms can help to overcome challenges and surpass the profit-oriented logics of private companies to achieve access to healthy and safe water, adequate sanitation, and improved hygiene, especially for vulnerable populations. This finding has transferability to other megacities in emerging countries that are facing public–private partnership debates on the provision of clean water and sanitation for all.

**Keywords:** public–private partnerships; IWRM; WASH; 2030 Agenda; Rio de Janeiro Metropolitan Region (RMRJ); Sustainable Development Goals (SDGs)

## 1. Introduction

With a minimum of 10 million inhabitants, megacities are urban conurbations that attract people through pull factors, such as economic growth, job opportunities, concentrated infrastructure and services, social diversity, and innovation. Many of these megacities face complex issues related to water, wastewater, and related risks (floods and stormwaters, health problems, water shortages, pollution of aquatic environments and soils, etc.). Difficulties in urban water management can be increased by the climate crisis and other complexities that relate to the economic realities of vulnerable populations which characterize the demographic pattern of megacities in developing countries [1,2].

Designed and implemented to "take transformative steps to a sustainable and resilient path on Earth, leaving no one behind", the United Nations 2030 Agenda for Sustainable Development (2030 Agenda) has established "prosperity for all" as one of its core principles [3]. However, no inclusive prosperity and opportunities for all can be envisioned without sustainable water and wastewater management in megacities, which accounted for approximately 13% of the world's total population in 2018 [4]. The proportion of the

world's population in megacities will continue to grow; even the current pandemic will not slow the trend toward urbanization across the planet. Aspirations for a better life, education for children, and decreased economic activities in rural areas remain the "push and pull" factors behind rural-to-urban migration. However, the urban experience is not universally better for all migrants: growing levels of inequality and exclusion remain "persistent trends in urban areas" [5].

The UN Report on The World's Cities in 2018 identified 33 megacities around the world, 26 of which are located in developing countries, with two in Brazil: São Paulo and Rio de Janeiro [4]. According to projections, 10 more cities will enter this category by 2030, all located in the world's least developed countries (LDCs) [2]. The UN points to fundamental linkages between the New Urban Agenda and the 2030 Agenda and estimates that the implementation of the urban dimension of the SDGs will cost US$38 trillion in global terms [5]. To address the issues outlined in these agendas, adequate funding, inter-institutional integration, and innovations in urban planning are vital to create sustainable megacities strategies in order to accomplish all 17 SDG goals, specifically Sustainable Development Goal 11 (SDG 11)—to make cities and human settlements inclusive, safe, resilient, and sustainable. From a climate crisis perspective, SDG 13 (take urgent action to combat climate change) must also be considered as fundamental, and action on SDG 17 (the need for cross-sector and cross-regional collaboration in pursuit of all the goals by the year 2030) must also be taken into consideration. From a human need perspective, as noted previously, Goal 6 (clean water and sanitation for all) remains foundational.

Based on the materials from UNESCO's *The Megacities Alliance for Water and Climate*, other technical reports, as well as the papers presented at the Water, Megacities and Climate Change Pre-Conference, this paper reviews the issues relating to water and wastewater opportunities and challenges in megacities around the world. This paper also presents key lessons from the Rio de Janeiro Metropolitan Region (RMRJ) of Brazil that can enlighten these discussions. From a general overview of challenges in megacities regarding clean water access, water security, climate change, and other vulnerability issues, a critical diagnosis of water and wastewater management in RMRJ in the light of the new Brazilian sanitation framework is attempted. In addition, considering the centricity of sustainable water management that encompasses SDG 6's targets and its interactions with SDG 11, this paper analyzes opportunities relating to private concessions for providing universal access to clean water, sanitation, and health (WASH) in the case of Rio de Janeiro. This is the first article that debates new Brazilian sanitation regulations concerning the second largest megacity in Brazil, Rio de Janerio, focusing on private sector participation, challenges, and opportunities.

## 2. Materials and Methods

This review consists of a compilation from the material available on UNESCO sites for the Megacities Alliance for Water and Climate—MAWaC [6]; the reports and scientific papers from the online pre-conference "Water, Megacities and Global Change" which took place from 7–11 December 2020 [7]; official information on the Brazilian basic sanitation sector available online [8]; current and relevant academic publications; combined with Brazilian specific technical documents regarding the new Brazilian Sanitation Framework, water security issues, and private sanitation concessions in the RMRJ published in 2019 and 2020, respectively. From the aforementioned data sources, 30 pre-conference online presentations and 20 scientific papers were selected for analysis due to their relevance regarding water and sanitation issues in megacities. It is worthwhile mentioning that according to Brazilian legislation basic sanitation encompasses not only clean water and wastewater treatment, but also solid waste management and urban drainage (also referred to as storm water management).

The main question regarding water, sanitation, and hygiene that arises in the Brazilian case is: is it universally possible to meet sustainability goals in profit-oriented models? To clarify this query, we also synthesized lessons learned from the Global Report Initiative

(GRI), as well as papers available on the website of the Brazilian Observatory of Water, Sanitation, and Human Rights—ONDAS Brazil [9]. In addition to this document review, participatory observations were also taken into consideration: the Brazilian authors have been active participants in the Rio de Janeiro Watershed Committees Forum since 2017, and both virtually attended the second public hearing on private concession in Rio de Janeiro State (which took place on 6 June 2020 and is partially available online) [10]. Furthermore, in combination, the Brazilian authors attended 10 of the 14 pre-conference sessions.

The document synthesis was validated through open interviews conducted with experts from governmental sectors who served as key informants on water and wastewater services in the study area. One of the authors of the *Technical Report* of the Oswaldo Cruz National Health Foundation (FIOCRUZ) on the Brazilian National Development Bank (BNDES) modelling for the RMRJ private sanitation concession was chosen to better incorporate water rights in relation to health and sanitation perspectives in the manuscript. A second key informant is a professor and scholar who currently represents the Rio de Janeiro State Watersheds Committees Forum (FFCBH) in the National Water Resources Management Plan revision process, and who was chosen on account of his comprehensive academic and practical knowledge of water and sanitation issues in Brazil and because he represented the Rio de Janeiro State at the pre-conference "Water, Megacities and Global Change".

## 3. Corporate Sustainability and the 2030 Agenda: General Considerations

In January 2020, just prior to the COVID-19 pandemic outbreak, the Davos Manifesto from the World Economic Forum (WEF) Annual Meeting stressed the importance of stakeholder capitalism as a response to the economic, social, and environmental challenges across the planet. Economic pressures have risen since then, revealing an accelerated need for the transition to sustainable and inclusive capitalism driven by the pandemic [11]. According to the Climate Change and Sustainability Services (CCSS) report, the post-pandemic investment landscape seems to be on a path that values the environment, social considerations, and good governance (ESG) more than ever before [12]. This method of review and valuation was proposed in 2004 by the UN Global Pact in partnership with the World Bank [13]. At the time, the 2030 Agenda was not widely adopted, but ESG information for a wide range of companies is increasingly available and appears to be having an impact on investor decisions, pushing the global business sector to provide accurate data on their sustainability practices in management reports. The Paris Climate Agreement also appears to be having an impact on investor decisions.

In Brazil, the understanding and applicability of ESG criteria is beginning to spread among large companies. Reporting on ESG criteria is being used to indicate greater transparency and solidity and to speak to improved resilience amid the uncertainties in a changing world. Companies that adhere to the ESG criteria report on and seek to improve: environmental requirements, such as carbon emissions, water consumption, waste generation, and deforestation; social requirements, such as equity, working conditions, inclusion and diversity policies, security, and impacts on the community; in addition to governance criteria with the development of anti-corruption programs, political lobbying, the structure and diversity of councils and collegiate bodies, communication, and transparency [14].

Comparing ESG practices requires establishing metrics and other standardization procedures: the Global Reporting Initiative (GRI) model is currently one of the most complete and most broadly agreed upon methods. Its elaborate processes require the definition of indicators, engagement of the organization's stakeholders, and reflections on the main impacts. The organization identifies its significant impacts over a period of time (generally annually) and evaluates whether these have been positive, negative, or trending in a quantifiable direction. Areas of evaluation include the economy, the environment, and society. The organization then reports their results according to globally accepted standards. Ideally, this results in a common language and a means of evaluating outcomes across a wide range of businesses and organizations [15].

GRI standards interact with other standards and references used worldwide. Among them, we highlight the UN 2030 Agenda and the Carbon Disclosure Project (CDP). The CDP is an international non-profit organization that administers a global disclosure system for managing the environmental impacts of investors, companies, cities, states, and regions in various countries around the world [16]. The inclusion of the SDGs in the GRI sustainability reports is necessary so that companies which adopt their standards can effectively contribute to the 2030 Global Agenda by reducing or reversing poverty, inequality, and environmental imbalance across the world, while improving practices relating to human rights, labor, the environment, and anti-corruption.

Currently, many companies have incorporated the broad themes covered by the SDGs into their sustainability reports: indicators relating to climate change, water management, and working conditions are increasingly common. GRI reports provide regular communication with company stakeholders, but they are also crucial sources of information for building trust with investors and partners and aligning investment through transparency and responsibility. The inclusion of 2030 Agenda themes in the GRI reports provides an important internal stimulus for decision making regarding contribution to the SDGs at all levels of the companies, boosting innovation and better performance, in addition to attracting investments. Where these are successfully applied, the opportunities for the private sector in finding solutions to global challenges while generating new business opportunities becomes clear [15].

In numbers, the GRI standards are present in reports of about 82% of the 250 largest companies in the world (G250) and so far in more than 15,000 organizations that have generated more than 63,000 reports currently in the database [17]. About 40 nations worldwide regularly refer to GRI in their policies and an additional approximately 50 nations are considering the implementation of these practices [18].

Regarding the Brazilian water and sanitation sector, the company that operates in RMRJ—the State Company for Water and Sewage (CEDAE)—is present in the GRI database, with a single report published in 2017. In part, the report follows the GRI standard [17]. However, it lacks detailed information on water and sewage at the RMRJ. In the report, specifically in the section on the production of treated water, the water treatment plant (WTP) ETA Guandu is highlighted as the largest in the world, with an installed capacity of 45,000 liters per second. The company's investments to increase water supply are outlined in the document, and the report highlights that the quantified values of losses in the water distribution system were as high as 30.1%. There is no information on the percentage of service to the population in the RMRJ. The main sources of water cited in the report are the Guandu and Paraíba do Sul Rivers and a reference is made to the 2014–2015 water shortage crisis. There is information on the size of the supply network in RMRJ (16,000 km) and the number of wastewater sewage treatment plants (WWTP) in the RMRJ at that time (22), with the largest capacity belonging to WWTP ETE Alegria, with the capacity to treat 5000 L per second of sewage at RMRJ. ETE Alegria also has a water reuse plant that has revitalized the port area in the municipality of Rio de Janeiro. A detailed description of RMRJ water and sanitation issues are presented in Section 5.1 of this report.

The CDP's Water Security Report, now in its 10th edition, provides the largest set of information on corporate freshwater data in the world, based on data provided by some 5000 companies [19]. This environmental information is used by more than 650 institutional economic agents to make decisions regarding investments in the portfolios of companies worldwide. GRI and CDP co-published the *GRI 303: Water and Effluents 2018 Report* [18]. The document presents the correlation between GRI water and wastewater indicators with the items in the CDP Water Security questionnaire, thus enabling consistency and comparability between the water resources data in their environmental and sustainability dimensions. The intention was to improve the quality and accuracy of corporate reports. Despite the COVID-19 pandemic, 88 cities have been recognized on the CDP A List in 2020 for their efforts in building resilience to safeguard the planet, economy, and their citizens.

These cities are leading sustainability actions in the world and Rio de Janeiro was among them (https://www.cdp.net/en/cities/cities-scores, accessed on 6 December 2021).

## 4. Water- and Wastewater-Related Challenges in Megacities around the World: Paths to Reach the 2030 Agenda's Sustainable Development Goals 6, 11, and 13

Water is at the core of the 2030 Agenda for Sustainable Development, which calls for inclusive, safe, resilient, and sustainable cities (SDG 11) [3,12,20,21]. Achieving SDG 6's targets plays a key role in achieving prosperity, health, and sustainability in urban settlements. Defined by the Global Water Partnership as a process which promotes the coordinated development and management of water, land, and related resources without compromising vital ecosystems, Integrated Water Resource Management (IWRM) has the potential to be effective and to guarantee universal access to water supply and sanitation, but also to promote increased levels of water security. Efficient and effective water management will help to address other urban challenges, such as floods, droughts, pollution, and the impacts of the climate crisis (SDG 13).

Climate change-driven extreme weather events and their impacts on the water cycle are amplified in megacities, where intense surface heating, overexploitation of available water, and water pollution coexist with urban population growth and aging water, wastewater, and stormwater infrastructure [20–24]. Regarding the impacts of urban growth and climate changes on each megacity's water and sanitation services, UNESCO's 2020 report highlights the emergence of critical vulnerabilities: megacities concentrate populations, services, and goods, thus amplifying water-related risks [5]. Problems such as water scarcity, cleanliness, and non-uniformity of water distribution are critical for the urban poor, who often lack access to basic water and sanitation services, and are at greater risk of suffering the consequences of natural disasters due to their housing often being located in hazardous areas. Slums are characteristic of emerging countries' megacities [2,5]. Emerging countries not only contained the largest share of all megacities worldwide in 2017 but are also expected to add the largest number of megacities between 2017–2030 [1]. Overcoming challenges, creating opportunities, and developing innovative solutions depend on available funding and financial capacities [22–25], complicating even further the situations for the global poor [2,26–29]. Public financing is not readily available in many nations: this adds to the need for governments to call on private partners for support.

Aged infrastructure and predominantly impermeable surfaces in urban areas can generate floods, water pollution, and reduce groundwater recharge: these are primary issues that often need to be addressed at multiple governance levels [2,12,20,24–39] An additional factor not often discussed is corruption in the water and wastewater sector: a lack of transparency and non-existent citizen engagement are factors that enable corruption in the awarding of contracts, shortfalls in construction, and issues with infrastructure operations and maintenance. The table below addresses these issues and outlines how they are addressed in various locations.

An additional issue has been identified by youth who participated in the Pre-Conference events. As those most impacted by past decisions and dire projections for the future, youth at the conference developed their own Youth Declaration. Issues that were emphasized included the need for long-term over short-term thinking, with the contention that short-term thinking was due to a "lack of strong institutional structures, systemic issues, and a lack of integrated thinking in policy and planning". The Declaration highlighted "the right to access to water, including safe and reliable water supply, sanitation, and hygiene access to the sewage treatment network as the major challenges" for the future. The Declaration also pointed out to the need for a "holistic approach that transcends hard and technology-centric solutions". Regarding this, among other suggestions, their Declaration cites off-grid sewage networks and independent underground sewage tanks as potential options for implementation. In addition, the youth delegates pointed to the importance of addressing human/nature connections as a way to enable an envisioned future where "society can develop and co-exist sustainably with nature" [40]. This Declaration is both visionary and

encompassing and suggests a wide range of metrics and issues that should be considered when examining water, wastewater, and stormwater provision in the world's megacities.

When summarizing the work that is being carried out worldwide on metrics, new approaches, and visions for the future, it becomes apparent that developing non-traditional approaches linked to circular economies for water, wasterwater, and stormwater sectors is the path forward [41]. This pathway links climate change, sustainable cities, and sustainable water and wastewater managent (SDGs 13, 11, and 6) respectively, to other SDGs that address energy (SDG 7), economic growth (SDG 8), SDG 12 on sustainable consumption and production, SDG 14 on oceans, and SDG 15 on life on land [42,43].

## 5. Water and Sanitation in Brazil: The New Brazilian Sanitation Framework and the Rio de Janeiro Case

*5.1. Clean Water and Wastewater Management; Water Security, Climate Change, and Vulnerability Issues*

With continental dimensions and a territory divided into 27 states and 5570 municipalities, Brazil faces unique challenges in developing a universal approach to wastewater services. In 2020, the country had approximately 212 million inhabitants, of which 84.72% were located in urban areas (180 million). In the southeast region of Brazil, where the Metropolitan Region (RMRJ) is located, 93.14% of the regional population is urban.

The Brazilian basic sanitation model has historically provided for the coordinated action of federal, state and municipal governments. While federal and state spheres have control over the provision of water, according to Brazilian Water Law 9.433/97, municipalities hold jurisdiction over sanitation with responsibilities relating to the management of systems either directly through municipal companies or jointly with the state when services are provided by state-operated companies. Regarding wastewater and sewage treatment, the Federal Government establishes general guidelines and establishes financial availability policies for investments. Water issue governance models are complex and involve executive government stakeholders, public and private regulatory agencies, federal and state water councils, and deliberative local watershed committees, as shown in Figure 1 [44].

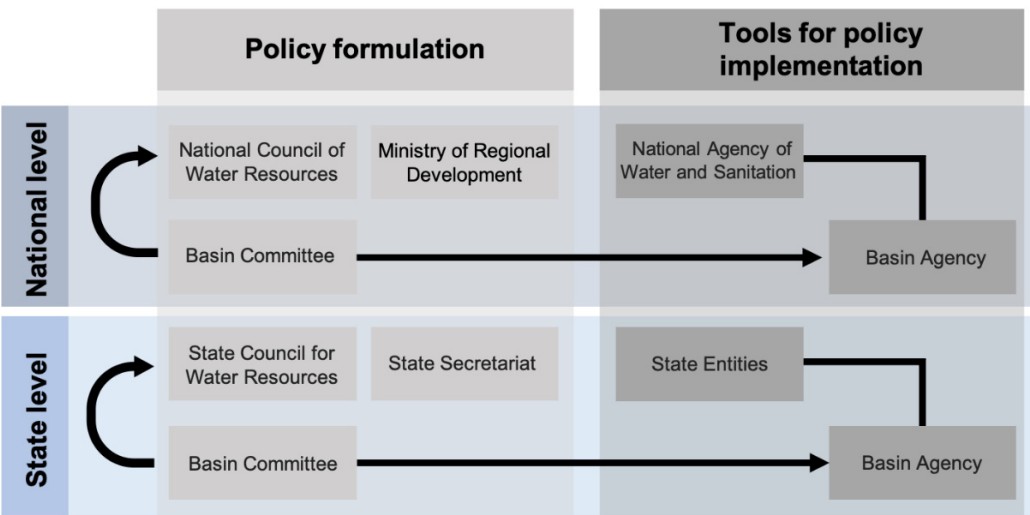

**Figure 1.** Brazilian water-management system. Adapted from MDR [44].

Enacted by Federal Law 14.026/2020, the new Brazilian sanitation framework broadened the National Water Agency's jurisdiction to encompass sanitation regulatory power in its broadest sense (water supply, wastewater and sewage treatment, urban drainage, and urban solid residues destination). The Ministry of Regional Development (MDR) established in 2019 is currently in charge of all sanitation management issues, taking IWRM away from the jurisdiction of the Environmental Ministry. Since 1995, the Brazilian government has gathered information on water and sewage services for the elaboration of a

huge diagnostics. The Diagnosis of Water and Sewage Services of the National Sanitation Information System (SNIS) is currently online [8] and provides annual information through a web system called SNISWeb, which is fed by state companies, municipalities, and private company self-declared data and is managed by the MDR.

In the SNIS-AE 2019 Report (for 2018 data), information was obtained from 2864 service providers that served 5191 municipalities with water supply services, corresponding to a resident urban population of 174.7 million inhabitants, which represents 93.2% of the total number of municipalities and 98.2% of the total urban population of Brazil. Regarding sanitary service assistance, 4226 municipalities were served, corresponding to 165.4 million inhabitants (75.9% in relation to the total number of municipalities and 92.9% in relation to the Brazilian urban population) [44].

In terms of water supply networks, in 2018, 680,400 km of infrastructure were serving 59.1 million water connections for 162.2 million inhabitants. Sewage collection infrastructure was smaller: there were 354,300 km of network leading to 34.6 million sewage connections. The water service index exhibited high values in the urban areas of Brazilian cities, with a national average of 92.9%, reaching up to 95.9% in the southeast. The rate of service by sewage networks reached about 108.1 million Brazilians (61.9% in urban areas of Brazilian cities), with emphasis on the southeast region, with an average of 83.7%. However, for sewage treatment, the service index reached 49.1% for the estimate of the sewage generated, corresponding to a volume of treated sewage of only 4.52 billion $m^3$ in 2019 [8,44].

With an urban population equal to 6,718,903 inhabitants, the Rio de Janeiro municipality has two sanitation service providers, CEDAE (a state company with regional coverage) and FABZO/RJ (a private company for local services). They cover 70% of Rio's urban area with so called adequate sanitation systems (4% of septic tanks, 66% of collect networks, and 30% untreated). Regarding average per capita water consumption, the state of Rio de Janeiro showed a value of 207.0 L per inhabitant per day in 2018, the highest among Brazilian states, 16.6% above the Southeast average, and 34.5% above the country average [44].

In terms of losses in water distribution, the Brazilian average reaches a worrying 39.2%, whereas in the southeast of Brazil this figure is slightly lower (36.1%) and in RMRJ it reaches 37.9% [44]. Losses in distribution are relevant, considering the current scenarios of water scarcity and high electricity costs, in addition to the increase in tariff costs and the overall waste of precious natural resources. In addition to the distribution losses, the apparent losses (measurement errors, clandestine connections, among others) are that the water is effectively consumed but is not billed by the service provider. The actual losses (physical losses) refer to leaks in pipelines, networks, branches, connections, reservoirs, and other operational units of the system, the latter being associated with the state of conservation of the pipes (materials used, age of the nets, preventive maintenance), the quality of the installation by the workmanship performed, and the existence of loss monitoring programs, among other factors [19].

The data available on the SNIS reflect the gap that still exists in Brazil for universal basic sanitation, especially regarding sewage collection and treatment, but on the other hand shows the great potential for privatization to the detriment of social assistance. In the case of the state of Rio de Janeiro, the Brazilian government points out that if the current operating conditions and investment capacity of the public water supply and sanitation services are maintained, it would take about 140 years to achieve universalization [10]. Despite this deficit, service providers in the sanitation sector in Brazil reached a total operating revenue of US$19.7 billion in 2018 versus total expenditure on services of US$ 17.1 billion, which shows the strength and economic potential of the sector. Among Brazilian concessionaires, CEDAE stands out in RMRJ, which presented the highest positive value for the relative difference between total operating revenue and total expense, being equal to 27.5%.

There is a clear privatization mandate within the new Brazilian Sanitation Law and its associated 10.588/2020 Federal Sanitation Decree. These documents have created controversy and serious concerns within technical and academic environments [45–50]. Official arguments that justify the new law speak to the country's lack of resources to

eliminate the significant deficits, leading to resort to resources from the private sector to solve these problems. Despite the widespread so-called efficiency of the private sector, international experience in the field of basic sanitation demonstrates inefficiencies with increased tariffs, reduced quality of service provision, and low levels of investments in new infrastructure, which has justified the resumption of public water and wastewater services in 311 cities and municipalities in 36 countries and solid waste in 85 cities in 11 countries, in different continents across the world [50–52].

Even if private capital is attracted to provide water and sewage services in Brazil, it is unlikely that the total water/wastewater deficit will be resolved due to uncertainties associated with the application of the new model. The impediment of inter-federal cooperation between municipalities and states in the provision of public basic sanitation services is in direct disregard of article 241 of the Brazilian Constitution, affecting municipal autonomy. Furthermore, the economic crisis stressed by the COVID-19 pandemic increases social pressures for the generalization of subsidies for poor populations through social tariffs in a context of the rigidity of new laws regarding contracts. New contracts signed now must set targets of 99% coverage for water supply and 90% for sewage collection and treatment by 2033.

RMRJ is inserted in three hydrographic regions (HR) of RJ: HR-II (Guandu Hydrographic Region), HR-V (Guanabara Bay Hydrographic Region), and HR-IV (Piabanha Hydrographic Region). About 17.5% of the municipality of Rio de Janeiro is inside the Guandu HR, while most of its area (82.5%) is located in the Guanabara Bay HR. The existing water sources in the two HR are strategic for the RMRJ water supply. Emphasis is put on the Guandu System, which receives a water transfer from the Paraíba do Sul River, with 60% of its waters coming through a complex system. According to the Regional Basic Sanitation Study of the Baixada Fluminense (2015), more than 12 million inhabitants are served by the waters of the Paraíba do Sul River Basin, fed through up to 160 m$^3$/s by the Santa Cecília pumping station. RH II and V have a total of 40 abstractions, in addition to the water transfer from the Paraíba do Sul River, which together supply the urban population of the RMRJ.

In the RMRJ, the state sanitation company has been fundamental to contributing to the improvement in service rates, even though there is still a lot that needs to be done to achieve universalization with quality services. The business model of Brazilian state sanitation companies was structured to establish cross-subsidies between municipalities, so that amounts collected in larger and richer municipalities are transferred to poorer municipalities with deficits, enabling appropriate operations and investments, that may otherwise not be achieved if they depended strictly on a local funding.

Article 2 of the Sanitation Decree reinforces regionalized provision, aiming to "generate scale gain" [53]. It states that existing public consortia and associated management arrangements and cooperation agreements can be recognized as regionalized provisions, so long as they do not cover municipalities in metropolitan regions and do not jeopardize economic and financial viability. In the absence of the formation of or adherence to regionalized structures, the municipalities will remain excluded from access to federal public resources. Comprising 22 municipalities within a total area of 6737.10 km$^2$, the Metropolitan Region of the State of Rio de Janeiro (RMRJ) was created in 1974 by Federal Complementary Law 20/74, which merged the States of Guanabara and Rio de Janeiro. According to State Complementary Law 184/2018, the municipalities that make up the RMRJ are: Rio de Janeiro, Belford Roxo, Cachoeiras de Macacu, Duque de Caxias, Guapimirim, Itaboraí, Itaguaí, Japeri, Magé, Maricá, Mesquita, Nilópolis, Niterói, Nova Iguaçu, Paracambi, Petrópolis, Queimados, Rio Bonito, São Gonçalo, São João de Meriti, Seropédica, and Tanguá (Figure 2).

The BNDES model for regionalization and private concession of CEDAE predicts service provision divided into four main regionalized blocks that cover Rio de Janeiro city and RMRJ, mixing municipal and metropolitan area territories with inner cities, as shown in Figure 3. In the modelled private concession, BNDES staff could not clarify why the future investments projected to be applied on the Guandu River watershed excluded

the municipality of Nova Iguaçu, which partially contributes to the Ipiranga River Basin, a tributary of the Guandu River that flows close to the catchment point of the Guandu Water Treatment Station (the biggest world WTTP, which serves more than 9 million people with potable water in RMRJ). According to FIOCRUZ´s technical note [50], the lack of priority for engineering and public health criteria related to vulnerable populations and the disregard for risk factors in the main water source of the RMRJ clearly emerges from the BNDES model. Urban water supply risks were evidenced in the beginning of 2020, with the presence of geosmin and 2-methyl-isoborneol (MIB) in the Guandu System waters, which generated an unprecedent water crisis in Rio de Janeiro. Many families, including low-income families, had to buy bottled water during the hot summer period due to improper water according to standards for human consumption.

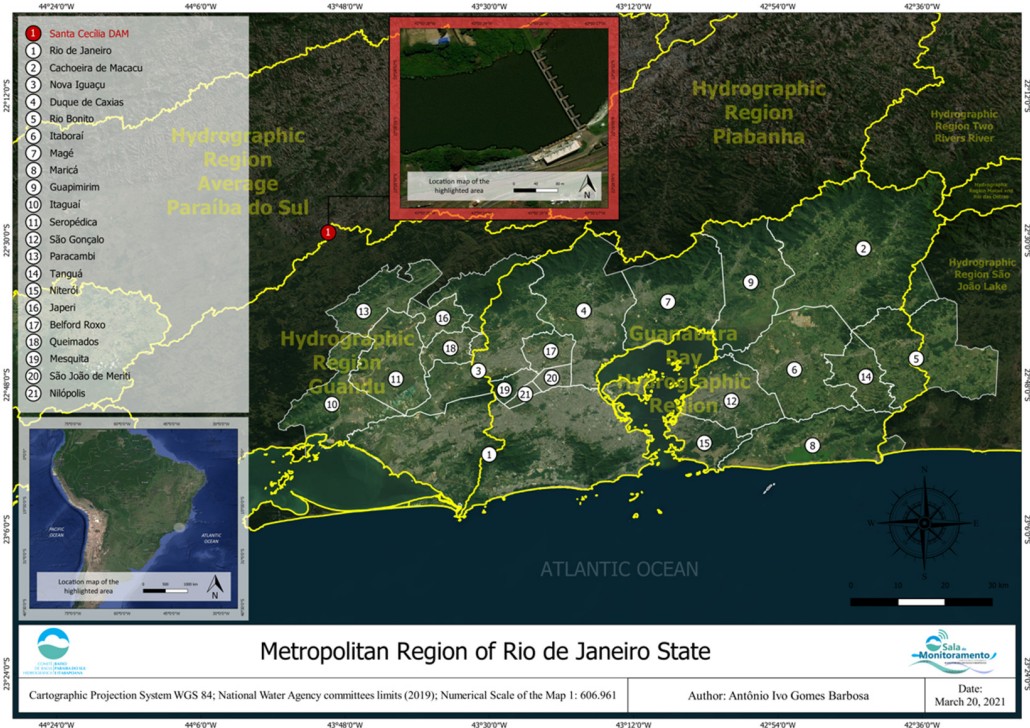

**Figure 2.** Rio de Janeiro Metropolitan Region and Rio de Janeiro State Hydrographic Regions.

The municipalities were grouped by the geographic, hydrographic, and operational economics of water supply and sewage systems, in addition to meeting market requirements, with a view to make the concession feasible. However, among the documents, studies, and plans made available on the public consultation page, no documents were uncovered that presents the study and the technical criteria that resulted in the proposed division [54]. Regarding the development and elaboration of the studies carried out for the modeling of the CEDAE's concession project, the FFCBH emphasized in its letter for BNDES that there was a lack of local representation previous to the public audience. This was due to the fact that participation of the municipalities involved was not guaranteed, neither for the local communities nor for the Watershed Committees that have jurisdiction of the hydrographic basins within the projected concession areas.

Similarly to FIOCRUZ, the FFCBH also noted that their BNDES model showed no identified concern with regard to the lack of articulation in the territory. It is understood that the guarantee of this articulation at the local level would contribute greatly to a better understanding of municipal realities at the hydrographic microbasin level, in order to provide greater effectiveness in proposing solutions and in planning services [50,54]. Both institutions revealed a concern that the planning proposed in the project will overlap with legally approved and valid instruments, such as municipal basic sanitation plans, rather than simply complementing one another.

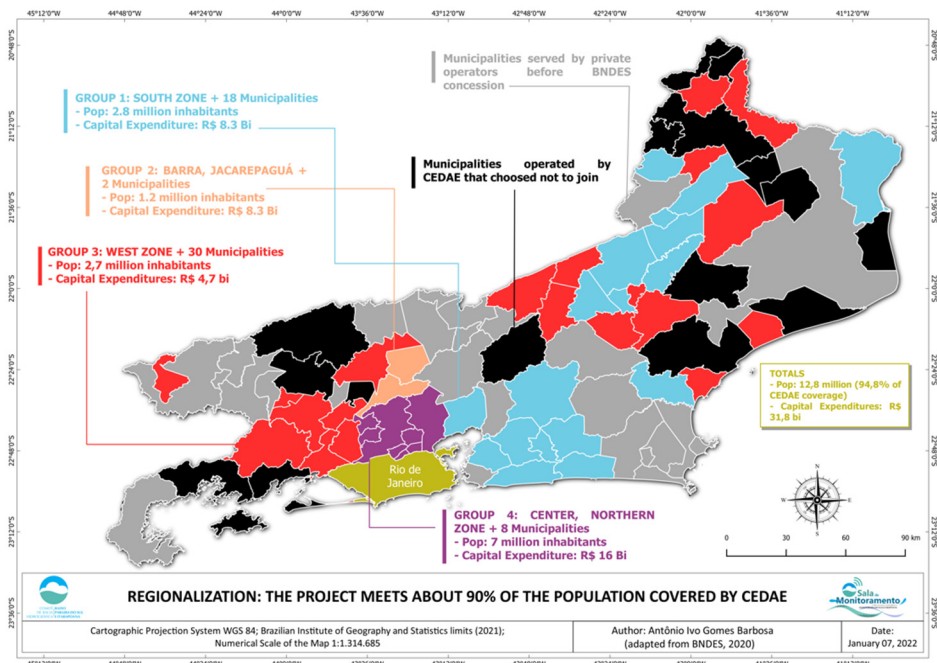

**Figure 3.** Blocks proposed by the National Development Bank for private concession of water and sanitation in Rio de Janeiro State.

The FFCBH pointed to the need for compatibilizing the planning of concession processes of services with river basin plans of the respective hydrographic regions. The Forum reiterated that compliance between watershed committees' water classifications and framings of water bodies, which are established according to water uses and quality standards, is not explicit in BNDES's model. In this sense, the need to consider these issues is emphasized, both in the planning documents and in the concession's contractual instruments. The role of the Regulatory Agency was not clear and the composition of the Monitoring Committee is fragile in terms of social issues. Thus, according to the FFCBH, all these issues bring uncertainty into the selection of sanitation solutions regarding not only the effectiveness of the sewage treatment, but also the guarantee for basic right to sanitation services.

### 5.2. Challenges, Opportunities, and Key Lessons for RMRJ in WASH Issues

Despite being on CDP's 2020 List A, Rio de Janeiro municipality also appeared on CDP's 2017 Cities Water Risk Map as having extremely serious magnitudinal risks. The risk description relates to leakages and illegal connections, and CDP pointed out a loss of around 40% of potable water supplied by the state company. As an adaptation action, besides investment in existing water supply infrastructure, the government of the state of Rio de Janeiro declared to CDP that CEDAE would invest in water metering and new technologies and personnel to extend network monitoring of and reduce response times to leaks, while also partially replacing the water distribution network and sewage collection system to reduce losses [55,56]. Regarding inadequate or aging infrastructures that appear as challenges on CDP 2020 data, adaptive action has evolved conservation awareness and education and watershed preservation [19].

It is important to emphasize that, by itself, the private concession process currently underway in the state of Rio de Janeiro does not necessarily imply greater operating efficiency in water systems nor the universalization of sanitation, as demanded by SDG 6. Recent academic literature and comprehensive studies point out that the policies linked to the reorganization in the provision of essential water and sanitation services (WSS) that has happened worldwide since the 1980s are intrinsic characteristics of the global expansion of certain forms of private-sector participation (PSP) in economic globalization dynamics [57,58].

Achieving SDG 6's targets in the RJMJ encompasses WASH challenges that are common in other cities mentioned in Table 1, ranging from socio-environmental to technical and managerial problems, all of which are interrelated. Social issues involve non-uniformity of water distribution and lack of access to basic water and sanitation services for people living in urban informal settlements, as well as urban poverty and inequality. The environmental dimension is linked to the radical alteration of natural hydrological processes, water scarcity, and water-related hazards (flood risk, drought, storm surge, and rising sea levels), difficulties regarding recharge of groundwater, the cleanliness of water bodies, and pollution of water supply sources. Technical and managerial issues relate to the fair provision of servicing, corruption, and viewpoints on the public good.

The majority of challenges cited in the Megacities Pre-Conference papers involve technical and managerial challenges in the RMRJ. As identified in the papers, weak and aging infrastructure, inadequate drainage networks, increases in impermeable surfaces, lack of integrated flood risk management systems, and poor management practices for extreme rain events have to be overcome. Weak or inefficient investment and the inadequacy of conventional management programs to restore water quality and improve the treatment quality of the wastewater treatment plants are also weaknesses in need of improvement. Costly wastewater treatment and remanent technical challenges, poorly developed water treatment networks, nonfunctional metered connections, violations of integrity, fraud and illegal water connections, rules governing recycled water use, and the integration of water/wastewater and land use policies are all managerial issues addressed in the Pre-Conference papers.

Urban storm water management is a recurring problem in RMRJ. The Paris Conference proposed solutions, including strengthening the connections of public and private stakeholders [35], while EU regulatory frameworks support private finance mechanisms as a means to promote NBS and hybrid water management solution projects [20]. Regarding private sector sustainability initiatives in Brazil, we highlight the B3's Corporate Sustainability Index (ISE B3), which was created in 2005: this represents the fourth corporate sustainability index created worldwide. The objective of the ISE B3 is to induce best ESG practices among the listed companies, comprising seven sustainability dimensions: (i) economic/financial; (ii) general; (iii) environmental; (iv) corporate governance; (v) social; (vi) climate change; and (vii) product nature. The diagnosis of companies' performance simulated by the ISE B3 application generates annual simulations. This allows the comparison of corporations' current portfolio performance levels and helps to fulfill stock market investors' needs, inducing private stakeholders to directly or indirectly finance water management actions [59].

The CDP 2020 database points to water security as one of the major issues threatening the Rio de Janeiro municipality. Achieving water security requires a holistic approach and a long-term perspective, involving water planning, allocation and pricing policies, and increasing water efficiency in industrial, agricultural, and domestic water uses. At the same time, it is necessary to ensure accessibility for vulnerable populations and promote favorable environments that support policies for the use of unconventional water sources. In 2019, the National Water Security Plan (PNSH) presented a careful analysis of water security levels throughout the Brazilian territory, defined through a Water Security Index (WSI-ISH). Applying the UN's water security definition, Brazilian ISH is structured in four dimensions: economic, ecosystem, resilience, and human well-being, considering socio-economic development, conservation of aquatic ecosystems, and drought and flood events. According to National Water and Sanitation Agency data, RMRJ's ISH ranges from low to minimum [45]. The state of Rio de Janeiro faces a critical situation due to water insecurity: 12,485,965 inhabitants (90% of its population), 69 million dollars (US) per year of its agricultural production (79%), and 17 billion dollars (US) per year (96%) of its industrial production is at risk [60].

**Table 1.** Challenges and opportunities regarding SDG 6- and SDG 11-related issues in megacities.

| SDG 6-/SDG 11-Related Issues | Challenges | Opportunities/ Remedial Measures | City(ies) | Reference |
|---|---|---|---|---|
| Interconnected water-related challenges | Understanding of long-term development to build resilient (port) cities and deltas | Applying geospatial mapping and innovative methodologies | Rotterdam, Hamburg, and London | [30] |
| Nature-based solutions (NBS) for urban water management | Radical alteration of natural hydrologic processes in megacities; Gray infrastructure systems which are vulnerable to intense rainfall events | Implementation of green infrastructure (GI) according to a Social–Ecological–Technological Systems Perspective (SETS) | Los Angeles, Chicago and New York City, USA | [31] |
| | Decision makers' limited views about the design and implementation of NBS and hybrid water management solutions | Higher mean land or property values in proximity to urban green space; Public and private finance attracted to NBS and hybrid water management solution projects | EU Regulatory Framework | [20] |
| | Coping with over-engineering and predominating impermeable surfaces | Implementation of ecohydrological NBS, hybrid system Integration of NBS and circular economy (CE) | Łódź, Poland | [24] |
| Preventive water management | Inadequacy of conventional management practices programs to restore water quality; Water protection before catchment | Governmental support system for innovative, modernized engineering articulated with agroenvironmental measures, payments for ecosystem services (PES), and agroforestry | Paris, France | [32] |
| Water resilience (risk management/flood/stormwater events/drought/pollution in climate change context) | Investment, innovation and integration of water and land use policies; Access to basic water and sanitation services for people living in urban informal settlements | GI implementation to (Green Infrastructure Strategic Plan) diversify sources of finance and promote PES | Sinking cities in coastal and delta regions (San Francisco, New Orleans, Hoboken, New Jersey, and Portland, in the USA; Amsterdam, The Netherlands; Brisbane in Australia; and Toronto in Canada) | [12] |
| | Water-related hazards (flood risk, drought, storm surge, and rising sea levels) | Identify the challenges with respect to financial constraints; Collaboratively developing, implementing, monitoring, and evaluating urban water resilience action plans | Cape Town, South Africa | [33] |
| | Weak infrastructure, inadequate drainage networks, lack of an integrated flood risk management system, and poverty | Build resilient homes: tree planting; floodwalls, raised entry points of houses, ditches dug to transport water | Port Harcourt and Lagos, Nigeria | [34] |
| | Recharge of groundwater; Management of extreme rain events | Integration of the stormwater question in urban planning; Connection of public and private stakeholders | Lyon, France | [35] |

**Table 1.** *Cont.*

| SDG 6-/SDG 11-Related Issues | Challenges | Opportunities/ Remedial Measures | City(ies) | Reference |
|---|---|---|---|---|
| Sanitation (wastewater management) | Improving the treatment quality of the wastewater treatment plants (WWTP); Changing wastewater dispatching practices in the sewer networks | Implementation of a real-time management of influent flowing through the networks | Ile-de-France (Paris conurbation) | [36] |
| | Source separation of urine and blackwater for nutrient recovery | Improved climate change impact and increase of avoided production of fertilizer | Belvédère, Bordeaux, France | [37] |
| | Costly wastewater treatment and remanent technical challenges | Improving packaging or collection systems for unused medicines; Improving wastewater treatment systems | France (surveys applied in Bourgogne) | [38] |
| | Incipient water treatment networks—less than 15% of the population is connected to a sewerage network and treatment plant; Pollution of water supply source (Saigon River) | Construction of new WWTPs; Improving nutrient processing technology in WWTPs and/or implementing alternative urban wastewater management (nature-based solutions) | Ho Chi Minh City, Vietnam | [39] |
| Urban water supply/groundwater (GW) management | Water scarcity, cleanliness, and non-uniformity of water distribution in informal slum settlements (ISS) | Composite system to expand and feed the supply system; Regular supply of tanker water during rainy season; Governmental help for gradual formalization of water networks | Mumbai, Lucknow and Kolkata, India | [27] |
| | Water scarcity and large dependence on groundwater; Non-revenue water; Increasing water abstraction through tube wells; Nonfunctional metered connections; Illegal water connections; Groundwater contamination/water pollution | Water audit; Reduction in water loss; Regularization of ilegal connections and unauthorized private tube wells; Strengthening the distribution network; Rain water harvesting structure; Roof top water structures; Minimizing contamination; Conserving surface water; Regulation on industrial water; Tariff system | Jaipur City, India | [28] |
| | Weakness of the investments made; Aging of the water supply network | Adequate investment extension of the water captation, storage, and distribution infrastructures; Effective partnership between state and concessionaire | Abidjan–Grand Bassam route, Côte d'Ivoire | [22] |
| | Effective increase in the available water volume to vulnerable populations; Awareness of resource limits Empowerment of the stakeholders | Assessment of GW reserve, water balance and wastewater treatment through an innovative data platform focusing on knowledge, education, and governance in integrated water resources management (IWRM) | Sahel, North Africa | [29] |

**Table 1.** *Cont.*

| SDG 6-/SDG 11-Related Issues | Challenges | Opportunities/ Remedial Measures | City(ies) | Reference |
|---|---|---|---|---|
| Water security (urban water reuse/risk management) | Legal, political, technical, and managerial challenges; Costs associated with utility operations; Rules governing recycled water use | Water Reuse Action Plan; Strong drivers able to overcome legal and political obstacles; Water reuse projects that are less costly than other new supply options | USA in general (USEPA) | [25] |
| | Vulnerability status of RMBH's public supply system due to multiple stressors | Implementation of risk management policy for public water supply | Brumadinho (Belo Horizonte Metropolitan Region—RMBH), Brazil | [39] |
| Corruption in water sector | Violations of integrity, fraud, and corruption that result in reduced quality, affordability, and availability of water and sanitation services | Applying the Water and Sanitation Integrity Risk Index (WIRI) based on three pillars: Investment Integrity, Operations Integrity, and Interactions Integrity | Asunción/Gran Asunción, Paraguay; Montevideo, Uruguay; Batumi and Tibisili, Georgia; Bucharest, Cluj, and Iasi, Romania; Budapest, Győr, and Nyíregyháza, Hungary; Kampala, Uganda; Nairobi, Kenya | [40] |

To overcome those challenges, the Rio de Janeiro State Water Safety Program (PROSEGH) was launched in 2021 (State Decree 47.498/2021). The general objective of PROSEGH is to establish integrated public strategies and actions that aim to reduce water vulnerability and ensure the availability of water, both in terms of quantity and quality, for human, environmental, and economic needs. Its main objectives are: to promote the integration of water resources management with other sectoral policies; increase investment synergy and efficiency in the implementation of actions and projects related to water security; guarantee surface and underground water supplies to current and future generations; minimize water vulnerability related to floods, droughts, and pollution; promote the protection, the conservation, and the recovery of sensitive areas; improve the environmental quality of water bodies and hydrographic basins; promote environmentally sustainable economic development; and strengthen educational actions linked to efficient and effective uses of water resources. To reach these goals, PROSEGH is structured in a four-axis model: planning, water supply, environmental quality, and water-associated risks management. In addition, the nation has added new regulations for non-potable water reuse (State Decree 47.403/20) and created the Technical Chamber of Green Infrastructure of the Rio de Janeiro State Water Resources Council (CTIV-CERHI).

Green infrastructures and nature-based solutions (NBS) were highlighted at COP 26 as two of the major strategies for fighting climate change, enhancing ecosystem services (ES) and promoting sustainability, thus overcoming the urban challenges (UC) described herein. Defined by the European Commission (EC) as "solutions that are inspired and supported by nature, which are cost-effective, simultaneously provide environmental, social and economic benefits and help build resilience," NBS are fundamental to mitigate several water-management problems in megacities. According to recent studies on NBS–UC–ES nexuses, "Green roofs, woodland-like, urban grasslands and meadows, horticultural gardens, (natural(ised) wetlands and natural(ised) ponds have links to a higher number of ES classes than other NBS [ . . . ], provide multiple benefits and adequately address multiple challenges" [61], p. 15–18.

The issues facing Brazil and Rio de Janeiro State are common across other jurisdictions. Kumar (2018) presents a comparitive study on water quality in eight different cities in South and Southeast Asia through the use of a Water Evaluation and Planning (WEAP) numerical simulation tool and concludes that better information and decision making is critical in addressing immediate issues [62]. This contention is supported in numerous studies, including Torre et al.'s (2021) examination of the provision of adequate waterwater systems

in the Global South [63]. An interesting comparison of water issues in two megacities facing divergent circumstances in their achievement of SDG goal 6 is found in Geere et al. (2021) and their comparison of São Paulo and London [64]. Overall, the achievement of Goal 6 and related SDG goals is relevant to megacities worldwide.

## 6. Conclusions

Access to clean drinking water and adequate sanitation has long been recognized as a fundamental human right. In conjunction with this is the need for improved stormwater treatment systems: this is increasingly important in urban environments faced with increased population growth, the rise in impermeable surfaces, and the impacts of the climate crisis. As the world continues to urbanize and the number of megacities grows, the role of governments and private agencies in achieving these basic rights becomes more critical. Concerted, coordinated efforts are required at local, regional, and national levels to ensure that these basic rights remain a focus for government spending or for engaging with private sectors who will provide these services.

The case study of Rio de Janeiro highlights the complexities in addressing water, wastewater, and stormwater issues in one of the world's megacities. Certainly, there have been advances in the development of regulations and policies, and a great deal of effort has been expended in defining jurisdictional responsibilities, new means of working cooperatively across watersheds, and addressing system inefficiencies. However, the root of issues in this case study remains the role of government in providing funding and developing public systems and the ongoing reliance on the private sector to provide adequate services, maintain a level of profitability, and address social, economic, and environmental issues. That is, while an academic, theoretical understanding of sustainability is certainly possible and nations across the world have taken the UN's Sustainable Development Goals to heart, there remains a disconnect between recognition of what needs to be done and what is actually done.

To the question, is it possible to address universality in water, sanitation, and hygiene within a system that relies on the profit-oriented private sector in the provision of services? the short answer is yes, but with provisions. Certainly, levels of government in Brazil and within the state of Rio de Janeiro are working diligently to address issues relating to the provision of services by private industries while still meeting worldwide sustainability targets and address the social, environmental, and economic issues facing the residents of one of the world's megacities. The approach taken by the Rio de Janeiro State Water Safety Program attempts to achieve this by promoting integrated strategies and actions through a four-axis approach: planning, water supply, environmental quality, and water-associated risk management. While government structures, jurisdictional responsibilities, financial capabilities, and vulnerable populations differ across the world's mega-cities, the Rio De Janeiro Program provides a model for other jurisdictions seeking to reduce water vulnerability, ensure efficiencies in financial structures and infrastructure, and protect urban environments. The plan is a step in the right direction, but there is much work to be undertaken to ensure that basic human needs are being met. Despite all the efforts of private companies in ESG processes all over the world, private concession logics adopted by the Brazilian government do not seem to guarantee WASH for all in RMRJ.

**Author Contributions:** Conceptualization, M.I.P.F., V.d.P.S.d.O., P.S. and G.S.; methodology, M.I.P.F.; formal analysis, M.I.P.F. and V.d.P.S.d.O.; investigation, M.I.P.F. and V.d.P.S.d.O.; writing—original draft preparation, M.I.P.F., V.d.P.S.d.O. and P.S.; writing—review and editing P.S. and G.S.; funding acquisition, P.S. and G.S. All authors have read and agreed to the published version of the manuscript.

**Funding:** This research received no external funding.

**Data Availability Statement:** The bulk of the data (references) used in this paper can be retrieved from six different sources: (1) UNESCO sites of The Megacities Alliance for Water and Climate—MAWaC (https://en.unesco.org/mawac, last accessed on 29 March 2021); (2) the reports and scientific papers from the online pre-conference "Water, Megacities and Global Change" which took place

from 7–11 December 2020 (https://en.unesco.org/events/eaumega2021/preconference#agenda, last accessed on 29 March 2021); (3) official information on the Brazilian basic sanitation sector available at (http://www.snis.gov.br/diagnosticos, last accessed on 26 April 2021); (4) https://ondasbrasil.org/biblioteca/artigos-2 (last accessed on 16 April 2021); (5) https://www.youtube.com/watch?v=rcdkx8AxKf4 (last accessed on 21 November 2021); and (6) https://www.globalreporting.org/search/ (last accessed on 22 November 2021).

**Acknowledgments:** To Jose Paulo Soares de Azevedo (COPPE/UFRJ) and to Alexandre Pessoa Dias (FIOCRUZ) for their fruitful discussions, to Antonio Ivo Gomes Barbosa (AGEVAP) for support with adaptive mapping, and to Laura Clark (VIU) for supporting the design of the graphical abstract. Finally, to all the members of Watershed Committees Forum of Rio de Janeiro State (FFCBH) for contributions via personal communications.

**Conflicts of Interest:** The authors declare no conflict of interest.

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
