# Peer review of "The Private Sector as a Partner for SDG 6-Related Issues in Megacities: Opportunities and Challenges in Rio de Janeiro, Brazil"

_sustainability, doi:10.3390/su14031597_

Round 1

Reviewer 1 Report

1.There are the lack of explanation of the abbreviations of IWRM, WASH, DNDSA etc. at the first appearance in the manuscript.

2.There is a need to clarify the theoretical contribution and practical implication in conclusion section.

3.In the Materials and Methods Section, authors mentioned that “From the aforementioned data sources, 30 pre-conference online presentations, and 20 scientific papers were selected for analysis” How did authors choose these document as analysis material, and what’s the selection criteria? In addition, authors conducted an open interview with an author of a technical report, how did author avoid subjective tendencies when there is only one respondent?

4.There are lots of conference paper in references. Each conference only focuses on several specific issues. So, I suggest that author should add some journal paper to follow the academic frontiers in a wider area. Besides that, please check the format of references to fit Sustainability Journal requirements before submit revised manuscript.

Author Response

Dear reviewer, thank you so much for your thoughtful feedback. Please see our responses to your comments/suggestions. The updated manuscript that includes changes based on your feedback will be uploaded through the mdpi portal on or prior to January 13th, 2022. 

1.There are the lack of explanation of the abbreviations of IWRM, WASH, DNDSA etc. at the first appearance in the manuscript.

The first time IWRM and WASH appears is within the keywords section which should have no definition, but we can define IWRM the first time it appears in the paper. WASH is simply - clean water, sanitation and health, not sure if we can define it any differently. DNDSA did not appear in the manuscript seach.

2.There is a need to clarify the theoretical contribution and practical implication in conclusion section.

This will be revised in the updated manuscript to meet the reviewer’s comments.

3.In the Materials and Methods Section, authors mentioned that “From the aforementioned data sources, 30 pre-conference online presentations, and 20 scientific papers were selected for analysis” How did authors choose these document as analysis material, and what’s the selection criteria? In addition, authors conducted an open interview with an author of a technical report, how did author avoid subjective tendencies when there is only one respondent?

They were selected because they were the papers most relevant to the topic of sanitation (we can add this clarifying comment to the paper). The key informant was chosen because he was the Brazilian representative (Professor) at the conference, and has comprehensive knowledge of sanitation issues in Brazil (we will add clarifying text to the paper regarding this also).

4.There are lots of conference paper in references. Each conference only focuses on several specific issues. So, I suggest that author should add some journal paper to follow the academic frontiers in a wider area. Besides that, please check the format of references to fit Sustainability Journal requirements before submit revised manuscript.

Yes, we will add further references to support our manuscript from other relevant academic sources.

Reviewer 2 Report

Manuscript ID 1522133

This review paper attempts to frame the water sustainable management related opportunities and challenges in megacities around the world. The subject is within the Scope of the Journal.

Some suggested improvements are:

  1. The title of the paper doesn’t sound clear enough in the part of «private sector as a partner for SDG 6 related issues».
  2. The abstract should be more structured: two sentences for context or challenge, research question or objective, materials and methods, key findings, implications and perspectives.
  3. Research question and findings should be clarified in the Intro section.
  4. The title of table 1 should be more specific in terms of the content. This table needs to be revised to present the results more succinctly.
  5. It is always important to highlight the novelty in the Intro and to show it in the main part. The novelty is not clear. If you frame the water sustainable management related opportunities and challenges in megacities please show this result in a more structured way. An attempt to list is given in table 1.
  6. The presented figures do not meet the requirements. The text in the figures is not legible.

Author Response

Dear reviewer, thank you so much for your thoughtful feedback. Please see our responses to your comments/suggestions. The updated manuscript that includes changes based on your feedback will be uploaded through the mdpi portal on or prior to January 13th, 2022. 

1. The title of the paper doesn’t sound clear enough in the part of «private sector as a partner for SDG 6 related issues».

We would prefer not to change the title much, we would be willing to change to “The private sector as a partner for SDG 6-related issues in megacities…..” 

2. The abstract should be more structured: two sentences for context or challenge, research question or objective, materials and methods, key findings, implications and perspectives.

We will make substantial changes to the abstract. 

3. Research question and findings should be clarified in the Intro section.

Yes, this will be added. 

4. The title of table 1 should be more specific in terms of the content. This table needs to be revised to present the results more succinctly.

We will change the title to “challenges and opportunities regarding SDG 6 and 11-related issues in megacities”; however, the information within the table is essential for the manuscript. We will leave it to the publisher to format the table to fit better.

5. It is always important to highlight the novelty in the Intro and to show it in the main part. The novelty is not clear. If you frame the water sustainable management related opportunities and challenges in megacities please show this result in a more structured way. An attempt to list is given in table 1.

Agreed, though more difficult with a review paper, but we will highlight the novelty that we can in the introduction.

6. The presented figures do not meet the requirements. The text in the figures is not legible.

All the figures have been sent to the publisher as a separate attachment in high quality, they are just low quality in the draft. 

Round 2

Reviewer 2 Report

I would like to thank the authors for reviewing their manuscript. The quality of the article has improved and the authors have replied to all suggestions.